



# Peatland trees record strong and temporally stable hydroclimate information in tree-ring δ13C and δ18O

Karolina Janecka[1,2], Kerstin Treydte[2,3], Silvia Piccinelli[1,4], Loïc Francon[1,5], Marçal Argelich Ninot[2], Johannes Edvardsson[6], Christophe Corona[1,7], Veiko Lehsten[8,9], Markus Stoffel[1,10,11]

[1]Climate Change Impacts and Risks in the Anthropocene (C-CIA), Institute for Environmental Sciences, University of Geneva, 1205, Geneva, Switzerland
[2]Research Unit Forest Dynamics, Swiss Federal Institute for Forest, Snow and Landscape Research WSL, 8903, Birmensdorf, Switzerland
[3]Oeschger Centre for Climate Change Research, University of Bern, 3012, Bern, Switzerland
[4]Division of Environment, Math, Psychology, and Health, Franklin University, 6924, Lugano, Switzerland
[5]Department of Geography, University of Bonn, 53115, Bonn, Germany
[6]Laboratory for Wood Anatomy and Dendrochronology, Department of Geology, Lund University, 221 00, Lund, Sweden
[7]Université Grenoble-Alpes, LECA UMR UGA-USMB-CNRS, 5553, Grenoble, France
[8]Department of Physical Geography and Ecosystem Science, Lund University, 221 00, Lund, Sweden
[9]Department of Natural Science, Design and Sustainable Development, Mit Sweden University, 831 25, Östersund, Sweden
[10]Department of Earth Sciences, University of Geneva, 1205, Geneva, Switzerland
[11]Department F.-A. Forel for Environmental and Aquatic Sciences, University of Geneva, 1205, Geneva, Switzerland

*Correspondence to*: Karolina Janecka (karolina.janecka@unige.ch)

**Abstract.** Peatland trees are valuable archives of paleoclimatic information; however, gaps persist in understanding the relationships between tree growth, peatland hydrology, and hydroclimate variables. While previous research in peatlands has mainly focused on tree-ring widths (TRW), yielding inconclusive results, the potential of stable carbon (δ13C) and oxygen (δ18O) isotopes in tree rings remains unexplored. In this study, we develop TRW, δ13C, and δ18O chronologies of Scots pine trees located in a Swedish peatland and a reference site on bedrock with a mineral soil layer. We assess their responses to hydroclimate conditions and evaluate their potential for reconstructing hydroclimate variations. Our findings show significant differences in mean TRW and δ13C values between the peatland and reference sites. Moreover, while all three proxies exhibit uniform year-to-year variations across sites, we observe discrepancies in long-term trends, particularly in δ13C. Although the climate sensitivity of TRW is weak and non-homogenous, the δ13C and δ18O peatland and reference chronologies contain robust and consistent signals, with a maximum sensitivity to water table, precipitation, and vapor pressure deficit (VPD) variations during summer. Both δ13C and δ18O chronologies show stable relationships with three key hydroclimate variables over time. In conclusion, while TRWs from living peatland pines at our sites have limited potential to record high-frequency hydroclimate information, δ13C and δ18O chronologies can serve as excellent proxies for the reconstruction of past hydroclimate changes.



## 1 Introduction

Peatlands are significant carbon repositories (Gorham, 1991) and valuable archives of past hydroclimate shifts, as evidenced by various proxies (Chambers and Charman, 2004). Among these, tree rings from both living and subfossil peatland trees can provide unique insights into seasonal climate variability with dating accuracy to the calendar year (Edvardsson et al., 2016). Yet, understanding the complex interactions between climate, tree growth, and peatland hydrology, with the latter being rarely recorder, remains a challenging field of research. Numerous questions persist regarding the main factors influencing tree growth in peatlands (Ballesteros-Cánovas et al., 2022; Dinella et al., 2019; Edvardsson et al., 2015a; Linderholm et al., 2002; Smiljanić et al., 2014; Smiljanić and Wilmking, 2018). In general, tree growth in these environments is linked to the depth and fluctuations of the local water table, making hydrological conditions the main driving factor (Boggie, 1972; Smiljanić et al., 2014). However, rising air temperatures also affect peatland tree growth, both directly by influencing seasonal temperatures and indirectly by enhancing soil evapotranspiration, which reduces waterlogging and ultimately promotes tree growth (Boggie, 1972; Edvardsson et al., 2015b).

Numerous dendroclimatological studies, mostly in Northern Europe, have explored the influence of hydroclimate on peatland tree growth. These studies consistently report challenges with cross-dating, due to missing or wedging rings (Edvardsson et al., 2012a; Pilcher et al., 1995; Smiljanić et al., 2014; Wilmking et al., 2012), and weak, temporally unstable, or non-existent relationships between tree-ring width (TRW) and hydroclimate variables (Cedro and Lamentowicz, 2011; Lamentowicz et al., 2009; Linderholm, 2001; Linderholm et al., 2002). Some studies also evidence complex growth responses that reflect a multiannual synthesis of hydroclimate conditions (Edvardsson et al., 2015a; Edvardsson and Hansson, 2015; Smiljanić et al., 2014). Complex dynamics within peatland ecosystems, including lag and feedback effects, have been suggested as contributing factors to the inconsistent climate-tree growth relationships (Edvardsson and Hansson, 2015; Linderholm et al., 2002; Smiljanić et al., 2014).

To date, systematic dendroclimatological studies utilizing stable isotopes from peatland trees as an alternative to TRW have not been conducted. Unlike traditional tree-ring parameters, such as TRW or maximum latewood density (MXD), stable isotopes are less dependent on specific ecological conditions (Briffa et al., 2002; Saurer et al., 2008; Treydte et al., 2007). This independence allows the use of stable isotopes from various environments, including lowlands, where classical tree-ring parameters often struggle to capture significant climate signals (Cernusak and English, 2015; Hartl-Meier et al., 2015).

Stable carbon ($\delta^{13}$C) and oxygen ($\delta^{18}$O) isotopes in tree rings are valuable proxies reflecting the physiological response of plants to climate and other environmental variables (Gessler et al., 2014; McCarroll and Loader, 2004; Siegwolf et al., 2022). Tree-ring $\delta^{13}$C depends on factors affecting photosynthetic uptake of $CO_2$ and is mainly controlled by stomatal conductance and the rate of carboxylation during photosynthesis (Farquhar et al., 1989; Siegwolf et al., 2022). Such that, warm and dry conditions reduce stomatal conductance and discrimination against $^{13}$C, resulting in higher $\delta^{13}$C values (Saurer et al., 1995; Siegwolf et al., 2022).





Tree-ring $\delta^{18}O$ reflects the combined influence of the $\delta^{18}O$ signature of source water taken up through the roots, and the leaf water $\delta^{18}O$ signature, which is driven by stomatal response to atmospheric vapor pressure deficit (VPD), temperature, and humidity via leaf water $^{18}O$ enrichment. Transpiration regulates the uptake of source water via roots, often sourced from precipitation carrying a specific $\delta^{18}O$ signal linked to air mass temperature (Rozanski et al., 1992). The impact of precipitation on $\delta^{18}O$, varies based on changes in the quantity and isotopic composition of infiltrated water (Treydte et al., 2014), soil water evaporation, and groundwater presence (Ehleringer and Dawson, 1992). During warm, dry periods with sufficient soil water, intensified leaf-level evaporative processes can result in increased $\delta^{18}O$ values in tree rings (Gessler et al., 2014; Siegwolf et al., 2022).

Several dendroclimatological studies, conducted across Central and Northern Europe, have consistently reported positive correlations between $\delta^{13}C$ and/or $\delta^{18}O$ and summer temperature and/or VPD, as well as negative correlations with precipitation and/or moisture. However, these studies also reported variations in the strength and/or temporal stability of responses among sites (Esper et al., 2018; Hartl-Meier et al., 2015; Hilasvuori et al., 2009; Reynolds-Henne et al., 2007; Saurer et al., 1995, 2008; Seftigen et al., 2011; Treydte et al., 2007, 2024), potentially posing challenges for climate reconstructions, particularly when integrating chronologies from various sites, as they may also contain different long-term trends (Esper et al., 2018; Seftigen et al., 2011; Wilmking et al., 2020).

Given the research gaps in dendroclimatological studies on peatlands and the potential of stable isotopes as an alternative to TRW, we here developed TRW, $\delta^{13}C$, and $\delta^{18}O$ chronologies of Scots pine trees from a Swedish peatland and an adjacent mineral soil site. We assessed the responses of TRW, $\delta^{13}C$, and $\delta^{18}O$ to hydroclimate conditions, and evaluated their potential for reconstructing hydroclimate variations. Specifically, we aimed to investigate (i) inter-site differences and similarities in mean values, between chronologies and their long-term trends, (ii) their responses to different hydroclimatic variables such as water table, precipitation, VPD, maximum, minimum, and mean temperatures, (iii) and the temporal stability of the strongest hydroclimatic signals.

## 2 Material and Methods

### 2.1 Mycklemossen peatland

Mycklemossen, situated in the Southwestern part of Sweden (58°21'N 12°10'E, 80 m a.s.l.; Fig. 1), is a peatland characterized by a mix of wet low areas (hollows) dominated by *Sphagnum rubellum* and *Rhynchospora alba*, alongside raised intermediate areas (hummocks) formed by the tussock-building sedge *Eriophorum vaginatum*. The tussocks comprise drier upper layers of peat, allowing for the establishment of low shrubs such as *Calluna vulgaris* and vegetation resembling forests with Scots pine (*Pinus sylvestris* L.) trees (Kelly et al., 2021; White et al., 2023). In general, the thickness of the peat layer across the peatland varies, reaching around 5 meters in the center and decreasing to about 2 meters at the edge. The peatland does not feature a distinct discharge zone, known as a lagg fen, which is typically located at the edge of the peatland and often represents the wettest part of the system (Howie and Meerveld, 2011). Consequently, trees growing at the edge of the peatland neither have



their roots submerged in water nor come into significant contact with groundwater throughout the year. The only sources of
nutrients and water for the surface vegetation and trees rooted in the acrotelm of the peatland are rainfall and snow

The regional climate is characterized by mild winters and cool summers, with an average annual temperature of 6.8°C.
July is the warmest month, with an average temperature of 16.4°C, while February stands as the coldest (-1.9°C). Annual
precipitation averages 770 mm, peaking in autumn (83 mm in October) and reaching a minimum in winter and spring (44 mm
in February-April) according to data from the nearest monitoring station, Vänersborg, covering the 1960-2019 period.
Our sampling sites are located within the Skogaryd Research Catchment, which is included in SITES (Swedish Infrastructure
for Ecosystem Science) and ICOS-Sweden (Integrated Carbon Observation System).

## 2.2 Sampling design

Three sampling sites were established on the Mycklemossen peatland: one in the center (CEN), one at the edge (EDG), and a
reference site (REF) (Fig. 1). Although trees at the CEN and EDG sites grow on slightly elevated, drier hummocks, both sites
are still relatively moist due to the proximity of trees with water pools (Fig. 1). In contrast, the reference site is located on
bedrock and features a well-drained mineral soil layer of 10-30 cm thickness, classified as a dry site. The reference site is
assumed to be comparable to other, ecologically non-extreme, temperate sites documented in the literature (Treydte et al.,
2007, 2024), where mixed climate signals are recorded in TRW. This comparison allows us to test whether site conditions
modulate TRW, $\delta^{13}$C, and $\delta^{18}$O variability and their responses to various hydroclimate variables. This is particularly important
for potential hydroclimate reconstruction, drawn not only from living peatland trees with known location but especially
subfossil trees, which might have been relocated from their original growth sites (Eckstein et al., 2009; Edvardsson et al., 2012,
2014). This poses potential challenges for hydroclimate reconstructions when wood originates from multiple locations
(Wilmking et al., 2020). Moreover, it is plausible that trees preserved in peatlands did not originally grow on the surface; for
instance, trees falling into the margins of developing peatlands may become preserved within the peat (Edvardsson et al.,
120    2016).

Given the challenges associated with cross-dating peatland tree rings, we collected two samples per tree from Scots
pine trees at each site (CEN: 20, EDG: 20, REF: 40 trees) using a 5-mm increment borer. The sampled trees showed no visible
signs of disturbances such as top-kill, scars, or wood rot. Trees in peatland ecosystems often experience instability due to high
water table levels, leading to leaning and the formation of compression wood. Although we observed weak compression wood
in some rings of our samples, previous research (Janecka et al., 2020) has shown that this has a minor impact on the climate
signal in tree-ring stable isotopes when cellulose is extracted. Consequently, we retained these samples for our analyses.



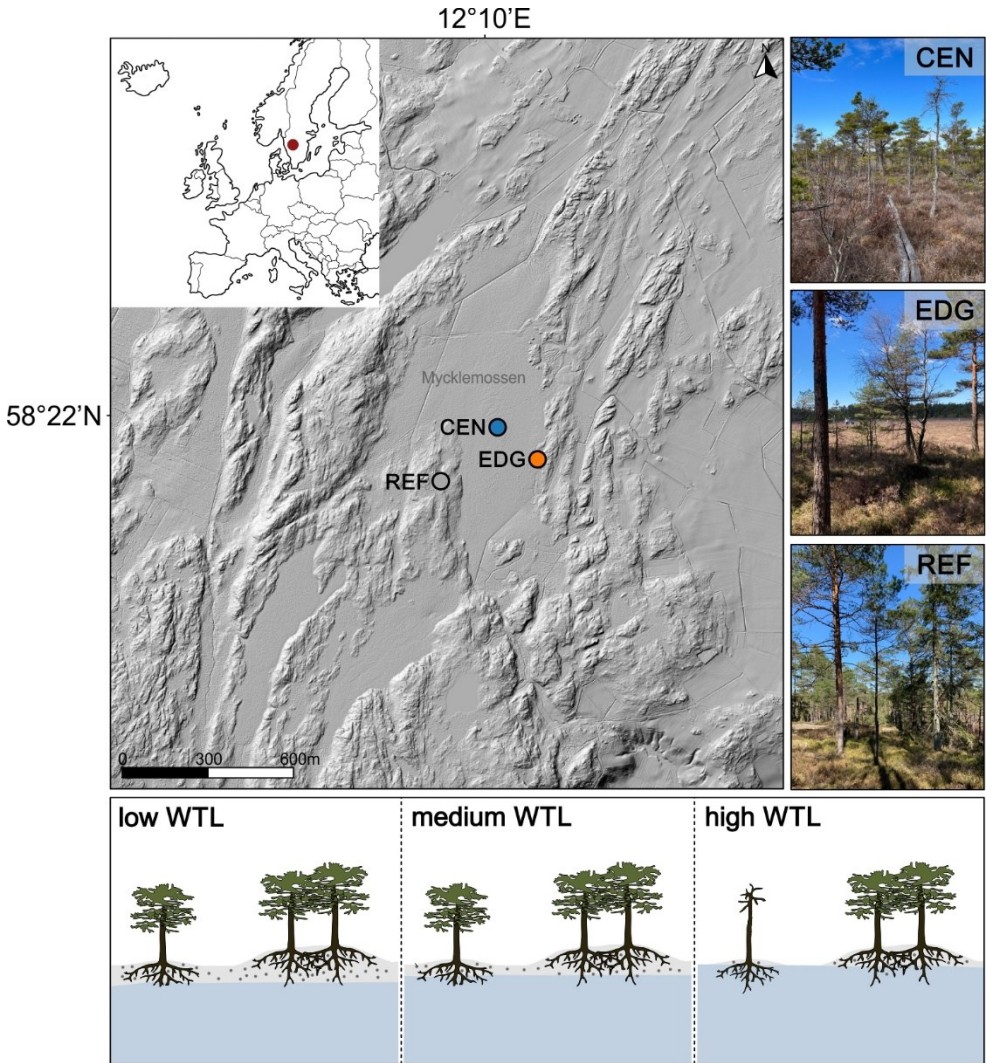

**Figure 1: Location of the study sites on the Mycklemossen peatland, Sweden (CEN = center, EDG = edge, REF = reference). The red dot on the inset map shows the location of the study area. The figure includes images of the center, edge, and reference sites. Source: ©Lantmäteriet (main image with relief shading); own collection (site-specific images). At the bottom, simplified illustrations show trees growing at the Mycklemossen peatland and the hypothetical access of their roots to water pools (blue shading) during low, medium, and high water-table level (WTL). During high WTL, trees growing on a larger hummock (bigger gray-shaded hill) are protected from flooding compared to those on smaller hummocks (smaller gray-shaded hill; own design; not to scale).**

## 2.3 Sample preparation and data treatment

First, all cores were cut with a microtome (Gärtner and Nievergelt, 2010), then slightly sanded with a very fine (2000-grit) sanding paper to enhance the visibility of the ring boundaries. The prepared samples were then scanned using a high-resolution scanner (1200 dpi, Epson Expression 10000XL). Tree-ring widths were measured from the resulting high-resolution images



using CooRecorder 9.3 (Larsson, 2013). To assign each ring to a calendar year, TRWs were visually crossdated and statistically verified using CDendro (Cybis Electronik & Data, Sweden; Larsson, 2013) and COFECHA (Holmes et al., 1986), respectively.

Following cross-dating validation, ensuring that no rings were missing and clearly defined ring borders, five trees per site (one core per tree) were selected for $\delta^{13}C$ and $\delta^{18}O$ measurements.

For isotope measurements, tree rings from 1960 to 2021 were separated annually using a scalpel under a binocular. Each tree ring was then cut into small pieces and packed into F57 Fiber Bags (200μm porosity) (ANKOM Technology, Macedon, NY, USA). Cellulose was extracted from the samples using a modified Jayme–Wise Holocellulose Isolation method (Boettger

et al., 2007)and was homogenized with an ultrasonic processor UP200S (Hielscher Technology, Teltow, Germany) (Laumer et al., 2009). In total, 915 tree-ring cellulose samples (i.e., 61 tree rings/radius × 15 radii = 915 samples) were individually weighed (1 ± 0.1 mg) and packed into silver capsules for isotope ratio measurements.

The samples were converted to CO by thermal decomposition at 1420°C with a TC/EA (Pyrocube, Elementar, Hanau, Germany) and C and O isotope ratios simultaneously analysed on an IRMS (MAT 253, Thermo) with a precision of 0.1‰.

$\delta^{13}C$ series were corrected for changes in the atmospheric $\delta^{13}C$ value due to anthropogenic activities (Belmecheri and Lavergne, 2020). All subsequent analyses refer to the $CO_2$-corrected data, referred to as 'raw $\delta^{13}C$'.

### 2.4 Hydroclimate data

Gridded monthly climate data (0.5°×0.5° lat x long) for minimum, maximum and mean temperatures, actual vapor pressure and precipitation sums were obtained from the CRU TS4.06 gridded dataset (Harris et al., 2020). A comparison from the

closest gridpoint to our study sites with instrumental records from two nearby meteorological stations (Kroppefjäll and Såtenäs, both ~30 km from the sampling sites) for the overlapping period 1960-2021 revealed highly significant correlations for individual months: minimum temperature ($r = 0.90$–$0.98$), maximum temperature ($r = 0.76$–$0.99$), mean temperature ($r = 0.92$–$0.98$), and precipitation ($r = 0.72$–$0.92$). Additionally, similar relationships were observed between isotope chronologies, and both gridded and instrumental hydroclimate data (tested, not presented). Given these strong and consistent correlations, we

chose to use the CRU dataset for the isotope-hydroclimate analysis in this study.

The atmospheric vapor pressure deficit (VPD) was calculated as the difference between saturated vapor pressure (vp) and actual vapor pressure. While actual vp data were provided by CRU, saturated vp was calculated following Eq. (1) based on Murray (1967):

$$\text{saturated vapor pressure} = 6.11 * 10\text{\textasciicircum}(\frac{7.5*\text{temperature}}{237.3+\text{temperature}}) \tag{1}$$

To estimate the water table levels at Mycklemossen peatland, we utilized half-hourly water table depth measurements from the Skogaryd Research Catchment (SITE project; https://meta.fieldsites.se/resources/stations/Skogaryd) that were converted to daily averages covering the period from October 20, 2015, to December 31, 2022. Due to the limited span of this 7 year dataset, we employed the Menyanthes impulse response model (Von Asmuth et al., 2002, 2008) to estimate past water table levels. This statistical model fits daily precipitation and potential evapotranspiration data to water table depth



measurements capturing the nonlinear relationship between these daily inputs and water table fluctuations, similar to the approach described by Lehsten et al. (2011). For model input, we used daily minimum, mean, and maximum temperatures and daily precipitation sums from two nearby weather stations (Kroppefjäll-Granan and Såtenäs) maintained by the Swedish Meteorological and Hydrological Institute. Potential evapotranspiration was calculated from minimum and maximum temperatures using the Mcguinnes Bordne formulation, with the implementation of Dey (2024). However, both nearby weather stations experienced either a 10-year downtime or were relocated, making it impractical to rely on a single station. To address this, a delta t correction was applied to harmonize the data from both stations, resulting in a homogeneous weather dataset (r-square > 0.90 for all parameters during the overlapping period). Despite some uncertainties in the station weather data, the model explained 80.6% of the variance in water table levels over the 7-year period. The root mean square error for predicted water table levels during the measured period was 3.0 cm, indicating reasonable accuracy for the model's estimates.

**2.5 Statistical analyses**

To analyze the data, we employed several statistical tests. The strength of common variation between the tree-individual isotope time series was tested by calculating the mean inter-series correlation (rbar) and the expressed population signal (EPS) (Wigley et al., 1984). To account for potential juvenile or non-climatic (e.g., $CO_2$ increase) trends and their typical patterns in the records of the different tree-ring parameters, all individual $\delta^{13}C$ and $\delta^{18}O$ series were detrended using a 30-year cubic smoothing spline with a 50% frequency cut-off. TRW series were detrended using a negative exponential curve (Cook and Peters, 1981). While detrending TRW data is essential for climate-growth analysis (Fritts, 1978), there is ongoing debate over the necessity of detrending isotope series (Büntgen et al., 2021; Helama et al., 2015; Torbenson et al., 2022), as the impact on isotope-climate responses can vary significantly (Esper et al., 2018). Consequently, we used both non-detrended ("raw") and spline-detrended ("detrended") stable isotope time series for correlations with hydroclimate data focusing on the most robust agreements. In total, we developed 15 chronologies (3 x TRW, 3 x $\delta^{13}C$ raw, 3 x $\delta^{13}C$ detrended, 3 x $\delta^{18}O$ raw, 3 x $\delta^{18}O$ detrended) by averaging the corresponding individual time series using the biweight robust mean (dplR package in R; Bunn et al., 2012). Descriptive statistics were calculated for all raw and detrended TRW, $\delta^{13}C$, and $\delta^{18}O$ series over the common 1960–2021 period.

Our raw TRW data deviated from a normal distribution, whereas the isotope data conformed to a normal distribution as determined by the Shapiro-Wilk test - a characteristic often observed in isotope data (Treydte et al., 2024). Consequently, we used the Wilcoxon test for raw TRW and t-test for raw $\delta^{13}C$ and $\delta^{18}O$ data to compare mean values across the three study sites. Additionally, we used Pearson's correlation coefficient to quantify the relationships between site-specific detrended TRW and raw $\delta^{13}C$ and $\delta^{18}O$ chronologies including their low frequency components. Visual and statistical comparisons between tree-ring parameters were conducted by calculating Pearson's correlation coefficients between detrended TRW and raw $\delta^{13}C$ and $\delta^{18}O$ chronologies, as well as their long-term trends.



To investigate the relationships between TRW, $\delta^{13}C$ and $\delta^{18}O$ variations and hydroclimate conditions, we calculated bootstrapped Pearson's correlation coefficients between the six raw ($\delta^{13}C$ and $\delta^{18}O$) and nine detrended (TRW, $\delta^{13}C$ and $\delta^{18}O$) site chronologies and monthly hydroclimate variables (water table, precipitation and VPD, but also maximum, minimum, and mean temperatures) over the 61-year period from 1960 to 2021. The significance of the correlation coefficients was tested using a bootstrapping procedure with 1,000 iterations (treeclim package in R; Zang and Biondi, 2015). This analysis spanned 21 individual months, from March of the year before xylem cell formation to October of the current year as well as the June-August season ($\delta^{13}C$ and $\delta^{18}O$ only). Statistical significance was determined at $p < 0.05$.

Building on the static correlation outcomes, we identified the season with the most robust link between $\delta^{13}C$, $\delta^{18}O$ and hydroclimate conditions. To examine the stability of these relationships over time, we applied a bootstrapped correlation analysis using 31-year moving windows lagged by 1 year, over the 1960–2021 period with 1,000 iterations.

## 3 Results

### 3.1 Chronology characteristics

The oldest trees were sampled at REF, where the average tree age was 118 years, with a maximum age of 142 years and a minimum age of 92 years. Trees at CEN and EDG were of similar age, with average values of 77 and 83 years, respectively. The oldest and youngest trees at CEN were 127 and 53 years old, while those at EDG were 116 and 44 years old.

### 3.1.1 Tree-ring width

We observed weak to strong common variations among tree-individual raw (Fig. 2) and detrended TRW series, with low to very high values of Rbar and EPS (Table 1). Rbar values ranged from 0.13 to 0.69 (raw) and 0.21 to 0.43 (detrended). EPS values ranged from 0.67 to 0.97 (raw) and 0.79 to 0.94 (detrended). The lag-1 autocorrelation values (AR1; Table 1) of the raw TRW chronologies fluctuated between 0.60 to 0.73 ($p < 0.01$). After detrending, the Lag-1 autocorrelation across all TRW chronologies decreased, with values ranging from 0.17 to 0.31 ($p < 0.01$).

### 3.1.2 $\delta^{13}C$ and $\delta^{18}O$

Strong common variations were found between the tree-individual raw $\delta^{13}C$ and $\delta^{18}O$ series at all sites (Fig. 3a and Fig. 3b), with moderate to very high values of Rbar and EPS (Table 1). Overall, Rbar and EPS values were slightly higher for detrended than raw data. Rbar values ranged from 0.35 to 0.79 (raw) and 0.45 to 0.77 (detrended) for $\delta^{13}C$ and were even higher for $\delta^{18}O$ with a range from 0.74 to 0.84 (raw) and 0.80 to 0.85 (detrended). EPS values were also higher for $\delta^{18}O$ than $\delta^{13}C$, with values fluctuating between 0.73 to 0.95 (raw) and 0.80 to 0.94 (detrended) for $\delta^{13}C$ and 0.93 to 0.96 (raw) and 0. 95 to 0.97 (detrended) for $\delta^{18}O$. Lag-1 autocorrelation values in the raw isotope chronologies varied between 0.40 and 0.51 ($p < 0.01$) for $\delta^{13}C$ and between 0.40 to 0.49 ($p < 0.01$) for $\delta^{18}O$. After detrending, the Lag-1 autocorrelation across all isotope chronologies decreased, ranging from 0.05 (non-significant) to 0.21 ($p < 0.01$) for $\delta^{13}C$ and from 0.23 to 0.33 for $\delta^{18}O$ ($p < 0.01$).



**Table 1: Characteristics of the raw and detrended TRW, δ¹³C, and δ¹⁸O data from the center (CEN), edge (EDG), and reference (REF) sites.**

| | TRW | | | | | | δ¹³C | | | | | | δ¹⁸O | | | | | |
|---|---|---|---|---|---|---|---|---|---|---|---|---|---|---|---|---|---|---|
| | CEN | | EDG | | REF | | CEN | | EDG | | REF | | CEN | | EDG | | REF | |
| | raw | det | raw | det | raw | det | raw | det | raw | det | raw | det | raw | det | raw | det | raw | det |
| **Rbar** | .69 | .26 | .13 | .21 | .41 | .43 | .35 | .45 | .47 | .54 | .79 | .77 | .76 | .82 | .74 | .80 | .84 | .85 |
| **EPS** | .97 | .84 | .67 | .79 | .94 | .94 | .73 | .80 | .82 | .85 | .95 | .94 | .94 | .96 | .93 | .95 | .96 | .97 |
| **AR1** | .73 | .17 | .69 | .30 | .60 | .31 | .51 | .14 | .40 | .05 | .46 | .21 | .41 | .23 | .40 | .23 | .49 | .33 |

## 3.2 Comparison of site chronologies

### 3.2.1 Tree-ring width

The mean raw TRW values at the REF site were the lowest and significantly differed from those at the CEN and EDG sites of the Mycklemossen peatland ($p < 0.001$; see boxplots in Fig. 2). The highest values were recorded at CEN (1.68 mm). Comparison of the detrended TRW chronologies across the sites indicated weak to medium coherence. Pearson correlation analysis indicated low to medium correlations between the sites with $r$-values ranging from 0.39 ($p < 0.01$) to 0.55 ($p < 0.001$); see table in Fig. 2). The low-pass filtered detrended TRW site chronologies (i.e., 30-year splines) revealed some discrepancies, including a distinct negative trend at CEN, that was not present at EDG and REF (Fig. 2). The correlations between the site chronologies ranged from 0.68 ($p < 0.001$, CEN-EDG) to 0.92 ($p < 0.001$, CEN-REF) (see table in Fig. 2).



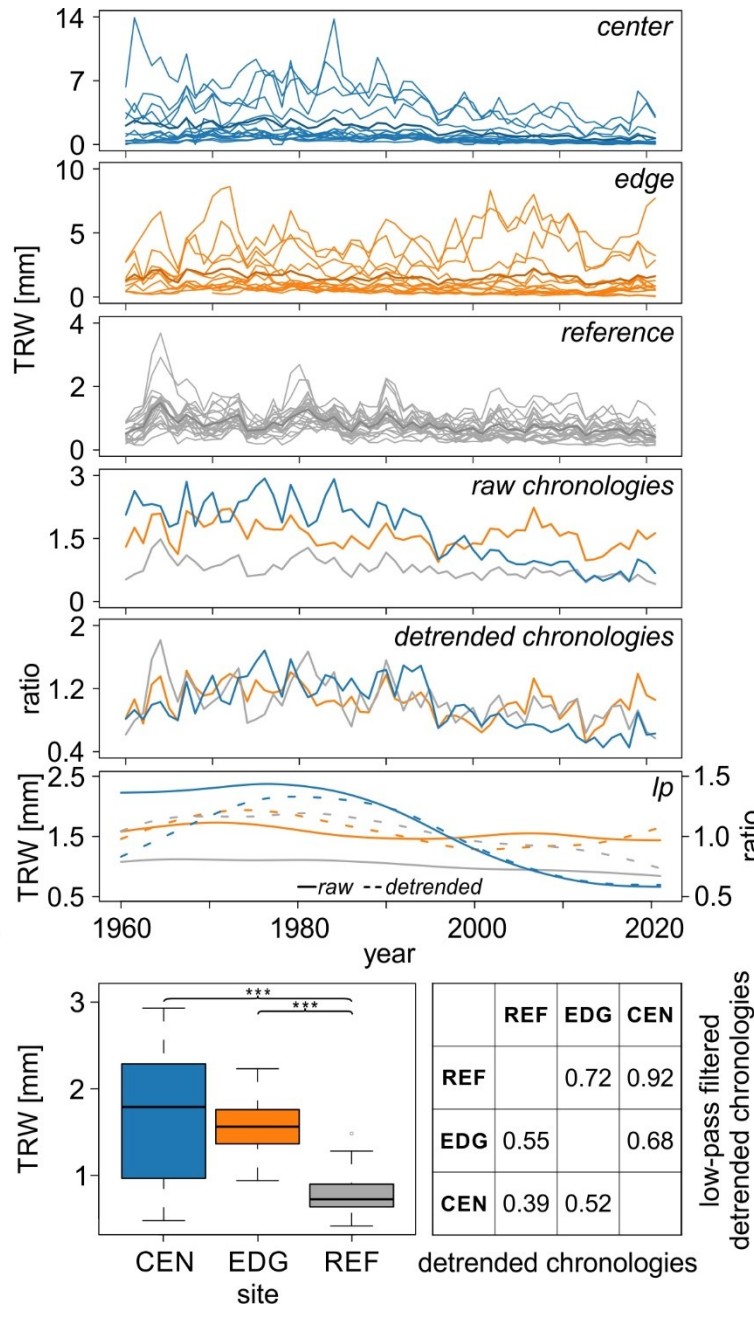

Figure 2: Tree-ring width (TRW) data spanning from 1960 to 2021 displayed for the center (CEN), edge (EDG), and reference (REF) sites; the raw and detrended chronologies; and the 30-year low-pass filters for both the raw (solid lines) and detrended (dashed lines) chronologies. The boxplots show the differences in mean raw TRW values across the three sites (*** p < 0.001). The table presents the correlations between detrended TRW site chronologies and 30-year low pass filters of the detrended TRW site chronologies.

## 3.2.2 δ¹³C and δ¹⁸O





Mean raw $\delta^{13}C$ values at REF significantly differed from the two peatland sites CEN and EDG (p < 0.001; boxplots in Fig. 3a). The highest $\delta^{13}C$ values were found at REF (-23.36‰), with the lowest recorded at CEN (-23.91‰). However, the mean raw $\delta^{13}C$ values of CEN and EDG did not differ significantly (boxplots in Fig. 3a). In contrast, the mean $\delta^{18}O$ values across all three sites, ranging from 27.80‰ at REF to 27.95‰ at EDG did not show statistically significant differences (boxplots in Fig. 3b).

Comparison of the raw isotope chronologies showed strong coherence between sites. Pearson correlation analysis indicated highly significant correlations between sites for both $\delta^{13}C$ and $\delta^{18}O$ (see table in Fig. 3a and 3b) with even stronger correlations for $\delta^{18}O$ (*r*-values between 0.86 and 0.93) compared to $\delta^{13}C$ (*r*-values between 0.40 and 0.76) (p < 0.001; see table in Fig. 3b and Fig. 3a, respectively).

          Low-pass filtered raw site chronologies (i.e., 30-year splines) showed some discrepancies with increasing trends in

$\delta^{13}C$ (Fig. 3a) and $\delta^{18}O$ (Fig. 3b) values over the most recent 15 years and more decadal variation in $\delta^{18}O$ (Fig. 3b) compared to $\delta^{13}C$ (Fig. 3a). The correlations between sites for $\delta^{13}C$ ranged from -0.06 (non-significant, CEN-EDG) to 0.95 (p < 0.001, EDG-REF) (see table in Fig. 3a) while for $\delta^{18}O$ they ranged from 0.41 (CEN-REF) to 0.77 (EDG-REF) (p < 0.001) (see table in Fig. 3b).







**Figure 3: Tree-ring (a) $\delta^{13}$C and (b) $\delta^{18}$O data spanning from 1960 to 2021. Each panel (a and b) presents the raw series from the center (CEN), edge (EDG), and reference (REF) sites; the corresponding raw and detrended chronologies; and 30-year low-pass filters of the raw (solid lines) and detrended (dashed lines) chronologies. Boxplots show the differences in mean raw $\delta^{13}$C and $\delta^{18}$O values across all three sites (*** p < 0.001). Tables display correlations between raw $\delta^{13}$C and $\delta^{18}$O site chronologies and 30-year low pass filters of the raw chronologies.**

## 3.3 Relationships between TRW, $\delta^{13}$C, and $\delta^{18}$O chronologies and their long-term trends

Comparisons between the TRW chronologies detrended using a negative exponential function and raw $\delta^{13}$C and $\delta^{18}$O chronologies indicated varying degrees of correlation, ranging from non-significant to highly significant (p < 0.001)



relationships (Fig. 4a). The correlations between the detrended TRW and raw $\delta^{13}$C chronologies ranged from -0.34 (REF; $p < 0.01$) to 0.46 (CEN; $p < 0.001$). Similar patterns were observed between the detrended TRW and raw $\delta^{18}$O chronologies, with values ranging from -0.04 (REF; non-significant) to 0.45 (CEN; $p < 0.001$).

The correlations between the low-frequency trends in detrended TRW and raw $\delta^{13}$C and $\delta^{18}$O chronologies varied from weak to strong (Fig. 4b). The values ranged from -0.40 (REF; $p < 0.05$) to 0.79 (CEN; $p < 0.001$) for TRW vs. $\delta^{13}$C and from -0.21 (REF; non-significant) to 0.79 (CEN; $p < 0.001$) for TRW vs. $\delta^{18}$O. Generally, the relationships between tree-ring parameters at the REF site were weak or non-significant, while at the CEN site, all correlations were significant.

.





**Figure 4. The relationship between (a) detrended TRW and raw $\delta^{13}C$ and $\delta^{18}O$ chronologies and (b) 30-year low pass filters of detrended TRW and raw isotope chronologies (*** p < 0.001, ** p < 0.01, * p < 0.05, ns = non-significant).**

## 3.4 Relationships between $\delta^{13}C$ and $\delta^{18}O$ chronologies and their long-term trends

Comparisons of the raw $\delta^{13}C$ and $\delta^{18}O$ chronologies indicated strong and significant (p < 0.001) relationships (Fig. 5a), with correlations ranging from 0.49 (EDG) to 0.54 (CEN). The low-frequency trends of the raw $\delta^{13}C$ and $\delta^{18}O$ chronologies were strongly correlated at CEN with $r$ = 0.93 (p < 0.001). However, differences were more marked at the REF site, where the



correlation was lower (r = 0.39, p < 0.01) (Fig. 5b). In general, the low-frequency domain of the raw $\delta^{18}$O chronologies showed more pronounced decadal variations compared to the $\delta^{13}$C chronologies.

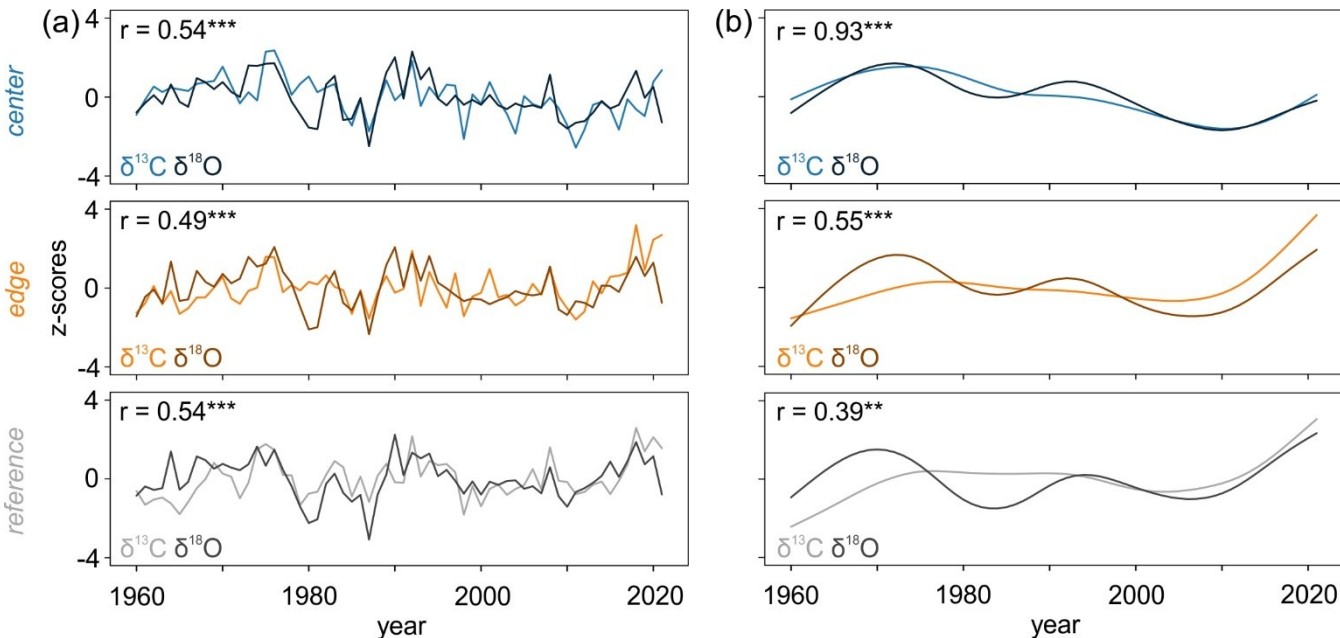

**Figure 5. The relationship between $\delta^{13}$C and $\delta^{18}$O (a) raw chronologies and (b) 30-year low pass filters of raw chronologies (*** p < 0.001, ** p < 0.01).**

### 3.5 Relationship of the TRW, $\delta^{13}$C, and $\delta^{18}$O chronologies to hydroclimate conditions

#### 3.5.1 Tree-ring width

The relationships between detrended TRW site chronologies and hydroclimate variables were generally weak, with no consistent response patterns emerging across the sites (Fig. 6). The most significant correlations were found for June of the year preceding xylem cell formation. Specifically, negative correlations were found between TRW and water table (with a maximum significant *r*-value of -0.38 at EDG) and precipitation (maximum significant *r*-value = -0.40 at CEN). Conversely, positive correlations were observed between TRW and VPD (maximum significant *r*-value = 0.42 at EDG) and temperatures

(maximum significant *r*-value of 0.36 for maximum temperature at EDG).



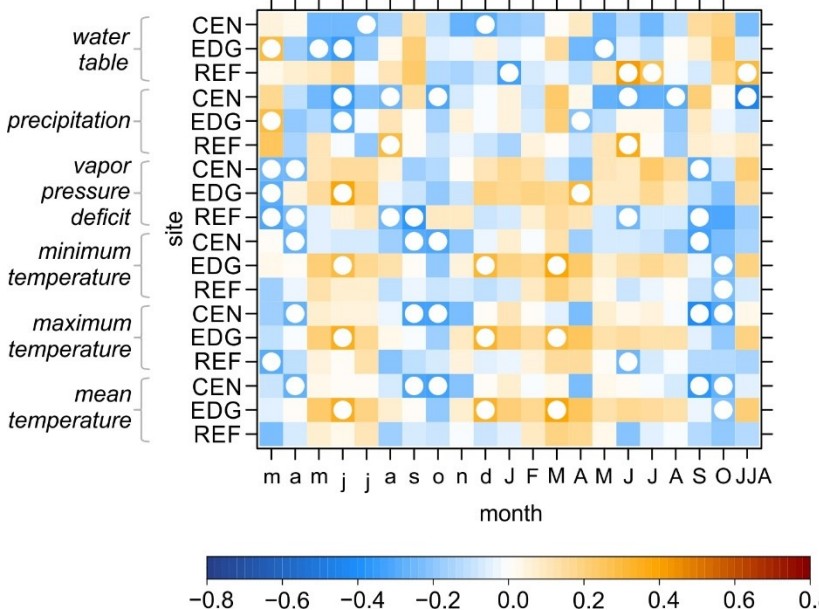

**Figure 6: Pearson correlation coefficients between the detrended tree-ring width site chronologies (CEN = center, EDG = edge, REF = reference) and hydroclimate variables. These variables include raw instrumental (water table) and raw gridded (precipitation, vapor pressure deficit, minimum, maximum, and mean temperatures) data. The correlations were computed for the period from March of the year preceding xylem cell formation to October of the current year and combined June-August (JJA). White circles indicate significant correlation at p < 0.05.**

### 3.5.2 $\delta^{13}C$ and $\delta^{18}O$

Strong and consistent relationships were observed between all $\delta^{13}C$ and $\delta^{18}O$ site chronologies and hydroclimate variables. Both raw and detrended isotope chronologies showed nearly identical climate signals (Fig. 7: raw; Fig. S1: detrended) and hence, only the results from the raw chronologies are presented here. Particularly strong correlations were found with variations of the water table, precipitation (negative correlations) and VPD (positive correlations) during the summer months (June to August) of the year in which the ring was formed (Fig. 7 and Fig. S2). Although all correlations were highly significant, ($p <$ 0.05), *r*-values were slightly higher for the $\delta^{13}C$ chronologies compared to the $\delta^{18}O$ chronologies. Correlations with minimum, maximum, and mean temperatures were slightly weaker in comparison to those with hydroclimatic variables. In addition, the seasonal response differed between $\delta^{13}C$ and $\delta^{18}O$: for the $\delta^{13}C$ chronologies, the strongest correlation remained most apparent during the summer months (June to August) particularly with maximum temperature whereas for $\delta^{18}O$, significant relationships shifted to the winter/early spring months (January to March) (Fig. 7).





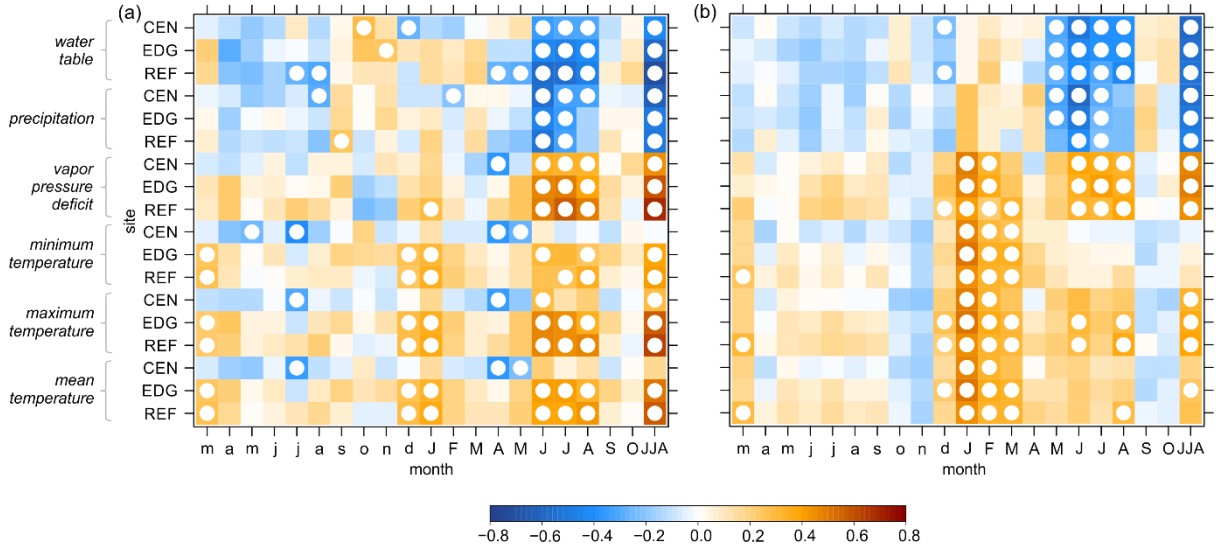

**Figure 7: Pearson correlation coefficients between the raw (a) $\delta^{13}$C and (b) $\delta^{18}$O site chronologies (CEN = center, EDG = edge, REF = reference) and hydroclimate variables. These variables include raw instrumental (water table) and raw gridded (precipitation, vapor pressure deficit, minimum, maximum, and mean temperatures) data. The correlations were computed for the period from March of the year preceding xylem cell formation to October of the current year and combined June-August (JJA). White circles indicate significant correlation at p < 0.05.**

## 3.6 Main hydroclimate drivers of $\delta^{13}$C and $\delta^{18}$O variations

Overall, the primary drivers of $\delta^{13}$C and $\delta^{18}$O variations across all three sites were the water table, precipitation, and VPD during the summer months (June to August) (Fig. 7 and Fig. 8). While the strength of the correlations varied between isotopes and sites, $\delta^{13}$C generally showed slightly higher correlations than $\delta^{18}$O, although the differences were not consistent. Specifically, $\delta^{13}$C showed increasing correlations from CEN to REF for water table and VPD, with the strongest precipitation signal found at CEN (Fig. 8a). Interestingly, although $\delta^{18}$O presented comparable correlations across sites, the two sites located within the peatland, i.e., CEN and EDG exhibited even stronger responses (p < 0.001) to all three hydroclimate variables compared to the REF site (Fig. 8b). The correlation values for $\delta^{13}$C chronologies ranged from -0.43 (CEN) to -0.72 (REF) for water table, -0.48 (EDG) to -0.63 (CEN) for precipitation, and 0.44 (CEN) to 0.68 (REF) for VPD (Fig. 8a). For $\delta^{18}$O chronologies, correlations ranged from -0.52 (REF) to -0.58 (CEN) for water table, -0.46 (REF) to -0.55 (CEN) for precipitation, and 0.44 (REF) to 0.49 (EDG) for VPD (Fig. 8b).





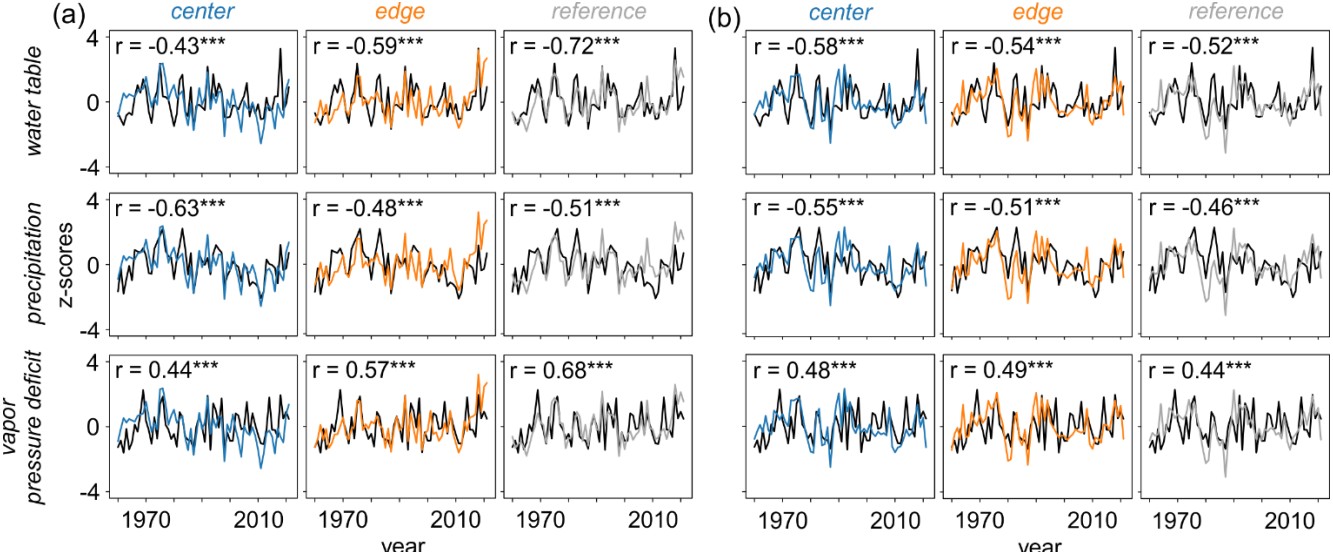

**Figure 8: Relationship between raw (a) δ¹³C and (b) δ¹⁸O site chronologies and June-August instrumental (water table) and gridded (precipitation and vapor pressure deficit) hydroclimate data (*** p < 0.001). Water table and precipitation data are presented with reversed y-axes for better clarity in interpreting the correlations.**

### 3.7 Temporal robustness of the summer hydroclimatic signals in δ¹³C and δ¹⁸O

Summer (JJA) of the current year of tree-ring formation was identified as the season with the highest correlations between the isotope chronologies and hydroclimate variables. To assess the temporal robustness of these signals, this period was selected for further analysis. Across all three sites significant relationships were consistently observed between δ¹³C (Fig. 9a) and δ¹⁸O (Fig. 9b) and water table, precipitation amount and VPD throughout the entire analysis period.

In addition, δ¹³C showed significant correlations with maximum and mean temperatures over time at all three sites, although there was a brief loss of sensitivity at the CEN site in recent years. The least stable and weakest, or even non-significant relationships were found between both isotopes and minimum temperature, as well as between δ¹⁸O and maximum and mean temperatures. Results slightly deviated for the detrended chronologies (Fig. S3) with minimum temperature becoming more significant for δ¹³C, whereas the water table and precipitation signals diminished in recent years.







**Figure 9: Temporal stability of the raw (a) δ¹³C and (b) δ¹⁸O site chronologies and hydroclimate data (water table, precipitation, vapor pressure deficit, minimum, maximum, and mean temperatures), calculated using a 31-year moving correlation. Solid lines indicate statistically significant correlations at p < 0.05**

**4 Discussion**

Our research offers a novel perspective on the complex interactions between climate, peatland hydrology, and tree growth. By examining not only tree-ring width, but also δ¹³C and δ¹⁸O chronologies of Scots pine in Southwestern Sweden, we provide new insights into peatland dendroclimatology. A key finding of our study is that the same hydroclimate variables influence both ¹³C/¹²C and ¹⁸O/¹⁶O fixation processes in tree rings. Furthermore, we find that while these variables significantly impact the isotopic signatures, local site conditions – such as the distinction between peatland and bedrock environments with a mineral soil layer – subtly affect δ¹³C and δ¹⁸O variability and their relationships with hydroclimatic conditions. Consistent





with previous studies on peatland pines, our findings show that the relationships between TRW and hydroclimate conditions are weak and inconsistent, making it difficult to pinpoint specific factors limiting tree growth.

## 4.1 Differences and similarities between site chronologies

### 4.1.1 Tree-ring width

Tree growth in peatlands differs from that on solid ground due to fluctuations in the water table as well as accessibility of tree roots to water and nutrients (Fig.1; Boggie, 1972; Eckstein et al., 2009; Edvardsson et al., 2012a). Elevated water tables reduce nutrient availability, hindering growth, while lower water tables generally promote radial growth (Fig. 1; Boggie, 1972; Penttilä, 1991; Freléchoux, et al., 2000; Vitas and Erlickytë, 2007). Similar to other European peatlands (Becker et al., 2008; Edvardsson et al., 2016; Holmgren et al., 2015; Smiljanić and Wilmking, 2018), trees on the Mycklemossen peatland grow on

slightly elevated, drier hummocks, which offer some protection from excessive flooding, while maintaining water access, typically fluctuating between 0 to -30 cm in June – August (Fig. 1; based on modelled data from 1960 to 2021). The presence of these hummocks and/or a lower water table likely contributes to wider TRW compared to the reference site (Fig. 2). However, the higher TRW variability at the peatland sites might have two explanations: either individual trees respond variably to drier conditions on hummocks (Carrer, 2011), or different hummock heights lead to differing growth conditions (Pouliot et

al., 2011). This variable growth highlights the complex interaction between regional and local climate, hydrology, and other factors, which drives peatland tree growth (Edvardsson et al., 2015a, 2016; Linderholm et al., 2002).

In contrast, the reference site, located on bedrock with a thin layer of highly permeable mineral soil, provides drier conditions for tree growth. Although the granitic bedrock can retain significant water volumes for trees (Nardini et al., 2024) due to its high porosity resulting from weathering processes (Migoń and Lidmar-Bergström, 2001), it still leads to reduced but more

synchronized tree growth compared to the peatland sites. The weak to moderate relationships observed between the site-specific chronologies (Fig. 2) suggest that while TRWs share a common signal, the latter is limited, reinforcing our earlier assumption that multiple co-existing factors influence tree growth across our sites (Edvardsson et al., 2015a, 2016; Linderholm et al., 2002).

The strong relationships in the long-term trends of the three TRW chronologies (Fig. 2) align with findings from

Lithuania (Edvardsson et al., 2015a) indicating that once peat soil becomes saturated with water, short-term changes in temperature or precipitation may not significantly alter the hydrological conditions that affect tree growth. Some studies (e.g., Gore and Goodall, 1983; Charman et al., 2004) suggest that hydrological responses in peatlands tend to be slow, driven primarily by persistent climate patterns rather than short-term fluctuations.

### 4.1.2 $\delta^{13}$C and $\delta^{18}$O

Significant difference in mean $\delta^{13}$C values among the sites, with the lowest values recorded at the peatland sites (Fig. 2) likely arise from variations in local site conditions. The peatland sites, although located on drier hummocks, still provide trees with





access to water pools (Fig. 1). In contrast, the reference site, located on bedrock, has a shallow mineral soil layer with higher water permeability. As mentioned before, although the granitic bedrock can retain substantial water for trees (Nardini et al., 2024), the growing conditions vary significantly between the peatland and reference sites, leading to noticeable differences in mean $\delta^{13}C$ values. The greater water availability at the peatland sites allows for higher stomatal conductance, resulting in lower $\delta^{13}C$ values compared to the reference site (Saurer et al., 1995; Siegwolf et al., 2022). One might expect that the generally higher water availability in peatlands would reduce stomatal aperture due to potentially anaerobic conditions (Kozlowski, 1997, 1984) resulting in less discrimination against $^{13}C$ and higher $\delta^{13}C$ values (Saurer et al., 1995; Siegwolf et al., 2022). However, as trees in the Mycklemossen peatland grow on hummocks, with convenient yet not excessive access to water pools (Fig. 1) and considering that their root systems tend to be shallow (>- 20 cm, Heikurainen, 1955; He et al., 2023) and spread horizontally (Edvardsson et al., 2012b); stomata can potentially remain open, leading to lower $\delta^{13}C$ values. In contrast, the drier conditions at the reference site lead to lower stomatal conductance. Higher $\delta^{13}C$ values highlight the dependence on soil moisture conditions (Leavitt, 1993; Saurer et al., 1995; Treydte et al., 2014), where the driest site records the highest $\delta^{13}C$ value, suggesting reduced stomatal conductance and higher water-use efficiency (Gessler et al., 2014; Saurer et al., 1995, 2014; Treydte et al., 2001).

The proximity of our sites excludes differences in the influence of precipitation $\delta^{18}O$ signatures and vapor pressure deficit on mean tree-ring $\delta^{18}O$ values, which exhibit remarkable similarity, as observed by Hartl-Meier et al. (2015) and Esper et al. (2018) at locally diverse (moist vs. dry) sites. Despite the ecological differences between our peatland and bedrock sites, the root systems at all locations are limited to the upper 0-30 centimetres of the soil layers, where only rainfall, including snow in winter, serves as the sole water source. It is widely acknowledged that soil water close to the surface is evaporatively enriched in $^{18}O$ compared with deeper soil water pools (Sarris et al., 2013; Treydte et al., 2014). This would suggest that trees at each location rely on $^{18}O$ enriched surface water, resulting in comparable tree-ring $\delta^{18}O$ values.

Although we documented strong similarities in the mean values and site-specific raw chronologies, particularly for $\delta^{18}O$ (Fig. 3b), long-term trends differ remarkably, especially for $\delta^{13}C$ (Fig. 3a). These discrepancies in $\delta^{13}C$ long-term trends, likely result from the complex interaction between climate dynamics and site-specific conditions. Specifically, the increase in air temperature due to climate change (Joelsson et al., 2023) likely enhances soil evapotranspiration differently across these ecologically diverse sites, creating slightly varied growth conditions (Li et al., 2014). The more positive long-term trend in $\delta^{13}C$ at the edge and reference sites over the past decade suggests that rising air temperature may have intensified soil water transpiration more at these sites compared to the center, leading to drier conditions, narrower stomata, and higher $\delta^{13}C$ values, particularly at the reference site. Conversely, the generally high-water levels at the center site may have buffered against the effects of rising air temperatures and intensive soil evapotranspiration leading to lower $\delta^{13}C$ values. The difference in long-term trends between peatland sites highlights the importance of careful examination of long-term trends in isotope data and avoiding the mixing of trees from different sites in climate reconstruction studies, as this could introduce artifacts that are difficult to statistically estimate (Esper et al., 2018).



### 4.3 Relationships between TRW, δ¹³C, and δ¹⁸O data

We analyzed the statistical relationships between detrended TRW and raw δ¹³C and ¹⁸O chronologies (Fig. 5a) as well as the longer-term trends extracted from these chronologies (Fig. 5b). Notably, we found strong positive correlations between these

430 chronologies, particularly at the central peatland site, where both TRW vs. δ¹³C and TRW vs. δ¹⁸O relationships were highly significant. The positive relationship between detrended TRW and raw δ¹³C chronologies at the central peatland site contrasts with the finding of Hartl-Meier et al. (2015), who reported a negative correlation, suggesting that dry conditions reduce carbon fixation and, consequently, suppress tree growth. Interestingly, our findings also indicate that a drop in water level reduces carbon fixation, resulting in higher δ¹³C values (Fig. 3a). However this reduction in water level simultaneously promotes tree

growth, a phenomenon typically observed when higher water table levels suppress growth (e.g., Edvardsson et al., 2016; Smiljanić and Wilmking, 2018). This suggests that under drier conditions, both in short- and long-term contexts, trees may benefit from increased nutrient availability in the soil (Freléchoux, et al., 2000; Penttilä, 1991). Additionally, as proposed by Edvardsson et al. (2015), trees may also utilize stored carbohydrates from previous years to produce wider rings further explaining the positive relationship between TRW and δ¹³C under these conditions.

Positive relationships between detrended TRW and raw δ¹⁸O chronologies suggest that lower water table levels and generally drier conditions on hummocks at the central peatland site may promote tree growth (e.g., Edvardsson et al., 2016; Smiljanić and Wilmking, 2018). This situation could be associated with stomatal closure, leading to lower δ¹⁸O values (Fig. 5a). However, we observed a contrasting response where enhanced tree growth was associated with higher δ¹⁸O values, indicating stomatal opening. This finding does not align with our previous interpretation of the interaction between TRW, δ¹³C,

and soil water conditions. Given these discrepancies and the inconsistent relationships - ranging from non-significant to weak and strong - between detrended TRW and raw stable isotope chronologies (Fig. 5a) as well as longer-term trends (Fig. 5b) from the edge site, we cannot definitively conclude that wood production is directly associated with stomatal conductance in peatland trees.

### 4.2 Relationships between δ¹³C and δ¹⁸O data

We evaluated the strength of the relationships between raw δ¹³C and δ¹⁸O chronologies (Fig. 6a) and their longer-term trends (Fig. 6b). Strong correlations between the δ¹³C and δ¹⁸O chronologies are consistent with findings from an isotope network across Europe (Treydte et al., 2007) supporting our hypothesis that these isotopic signatures are predominantly linked through leaf-level processes in the high-frequency domain (Treydte et al., 2007).

Interestingly, the relationships between longer-term trends show significant variability (Fig. 6b). The peatland site

located at the center show a notably high correlation while the edge and reference sites exhibit moderate to low correlations, respectively. Generally, the low correlations between δ¹³C and δ¹⁸O long-term trends can be attributed to at least two factors. First, δ¹³C records are influenced by tree size and stand dynamics, whereas δ¹⁸O records are less affected by these factors



(Klesse et al., 2018). Second, during the industrial period (20[th]/21[st] century), long-term trends in $\delta^{13}C$ records may reflect non-climatic influences such as changes in stomatal conductance in response to rising atmospheric $CO_2$ concentrations (Treydte et al., 2009). In contrast, $\delta^{18}O$ records tend to have minimal to no non-climatic trends (Young et al., 2011), preserving long-term climatic variation more reliably (Treydte et al., 2024). While combining $\delta^{13}C$ and $\delta^{18}O$ can enhance climate signal (Loader et al., 2008; Treydte et al., 2007) and previous (hydro)climate reconstructions have combined both $\delta^{13}C$ and $\delta^{18}O$ data (Bégin et al., 2015; Büntgen et al., 2021), it is imperative to individually examine the long-term trends of each isotope chronology before merging the data. This careful examination helps prevent the conflation of divergent long-term trends and ensures the integrity of climate reconstructions (Treydte et al., 2007).

### 4.4 Hydroclimate signals

#### 4.4.1 Tree-ring width

Our findings reveal weak and non-systematic (Fig. 6) relationships between detrended TRW chronologies and hydroclimate variables, a result consistent with other studies (e.g., Linderholm, 2001; Linderholm et al., 2002; Lamentowicz et al., 2009; Cedro and Lamentowicz, 2011; Edvardsson et al., 2015; Edvardsson and Hansson, 2015). Significant responses were mostly observed in June of the year preceding xylem cell formation and were evidenced only at the peatland sites. The limited and delayed responses are likely due to the reliance of peatland pines on fluctuations in the water table, in addition to direct effects of climate on growth (Linderholm et al., 2002).

Climate changes may not immediately affect the water table level which can take a few years to several decades to respond (Kilian et al., 1995). Our findings regarding the influence of past climate on peatland pine growth support the hypothesis of this lag, (Edvardsson and Hansson, 2015; Linderholm et al., 2002; Smiljanić et al., 2014) and the idea that TRWs in peatland pines may not be suitable for high frequency hydroclimate reconstructions (Linderholm et al., 2002).

#### 4.4.2 $\delta^{13}C$ and $\delta^{18}O$

We observed strong inter-site similarity between $\delta^{13}C$ and particularly $\delta^{18}O$ chronologies (Fig. 2) suggesting a common hydroclimate signal (Saurer et al., 2008). This is confirmed by our dendroecological analysis (Fig. 7), which shows that summer hydroclimate conditions, particularly water table, precipitation, and vapor pressure deficit (Fig. 8), are key drivers of $\delta^{13}C$ and $\delta^{18}O$ variations. These results are consistent with findings documented at ecologically diverse sites in Central and Northern Europe (Esper et al., 2018; Hartl-Meier et al., 2015; Saurer et al., 2008; Treydte et al., 2024). The minor relevance of previous-year conditions also align with other studies (Esper et al., 2018; Hartl-Meier et al., 2015; Reynolds-Henne et al., 2007; Seftigen et al., 2011; Treydte et al., 2024), indicating that carryover effects into subsequent tree-ring cellulose $\delta^{13}C$ and $\delta^{18}O$ are negligible. This suggests that if water transport is sufficient (Martínez-Sancho et al., 2023), the isotopic signature of tree-ring cellulose is predominantly created during the peak of meristematic activity and xylem cell-wall thickening in summer (Cuny et al., 2015; Treydte et al., 2024). This assumption is also confirmed by the dendrometer and wood phenological investigation



performed on the Mycklemossen peatland, in parallel to this isotope study (Francon et al., 2024). The positive correlations

between TRW and $\delta^{13}C$, suggesting that trees from the central peatland utilize stored carbohydrates to produce wider rings, may therefore be disregarded. Subtle yet consistent differences in the $\delta^{18}O$ response strength reveal that peatland sites exhibit higher sensitivity to key summer hydroclimate parameters compared to the reference site on bedrock (Fig. 8b). This suggests that $\delta^{18}O$ chronologies from peatland sites are more reliable proxy for past hydroclimate changes. Obviously, higher soil water availability in peatland likely facilitates stomatal opening and continued transpiration even under high VPD conditions

resulting in evaporative leaf water $^{18}O$ enrichment and higher isotopic values in the newly produced assimilates utilized for cellulose synthesis (Gessler et al., 2013, 2014; Offermann et al., 2011; Treydte et al., 2014). The weaker correlations at the reference site suggest less sensitivity to climate variables potentially due to more frequent stomatal closure and less variation in $\delta^{18}O$ in response to climate. This implies that under moist peatland conditions, the relative variations in leaf water enrichment due to increased transpiration outweigh the relative variations in soil water $\delta^{18}O$, leading to stronger associations between

hydroclimate and $\delta^{18}O$ at these locations (Hartl-Meier et al., 2015). Despite similar mean values across sites (Fig. 3), the stronger correlations between $\delta^{18}O$ and hydroclimate parameters at peatland sites and positive correlations between TRW and $\delta^{18}O$ (Fig. 4), suggest that soil moisture content and stomatal size are not the sole drivers of $\delta^{18}O$ variations. Simultaneously, we cannot exclude that trees at all sites solely utilize surface water enriched in $^{18}O$. These findings are consistent with those reported by Treydte et al. (2014), Hartl-Meier et al. (2015), and Esper et al. (2018), indicating that climate signals in tree-ring

$\delta^{18}O$ variations are most pronounced at temperate sites with humid conditions.

Additionally, we observed a positive impact of winter/early spring (January to March) temperatures on $\delta^{18}O$ (Fig. 7). This relationship could be attributed to snowmelt dynamics. Warmer temperatures can induce snowmelt or rainfall leading to isotopic enrichment in surface water due to evaporation and sublimation processes (Dahlke and Lyon, 2013) This isotopically enriched water can be incorporated into tree rings, resulting in higher $\delta^{18}O$ values. Warmer temperatures also accelerate

snowmelt, potentially reducing snowmelt water availability and increasing reliance on rainwater. Trees in peatlands and at the reference site, constrained by shallow root systems or bedrock (He et al., 2023; Heikurainen, 1955), may thus use isotopically enriched surface water, contributing to higher $\delta^{18}O$ values.

A distinct pattern, similar to the one observed for $\delta^{18}O$, is not evident in the $\delta^{13}C$ responses to hydroclimate parameters. While mean $\delta^{13}C$ values are consistently lower at the peatland sites compared to the reference site (Fig. 3), the

strength of responses to key hydroclimate parameters varies (Fig. 8). This suggests that the hydroclimate signal preserved in $\delta^{13}C$ is more susceptible to influences from differences in site conditions, resulting in greater variability in response strength. $\delta^{13}C$ appears to be a more sensitive indicator of environmental stress compared to $\delta^{18}O$, as variations in stomatal conductance are more pronounced at the reference site than at the peatland sites. Trees at the reference site must regulate transpiration more carefully to prevent excessive water loss which supports our previous finding that stomata tend to remain more closed at the

reference site. This results in higher $\delta^{13}C$ values (Fig. 3) and stronger relationships with the water table (Fig. 8).

We also observed stronger $\delta^{13}C$ responses to VPD at the reference site, suggesting that trees adjust their carbon assimilation processes in response to the combined effects of soil and atmospheric conditions (Fonti et al., 2013; Zeiger and Farquhar,





1987). Specifically, reduced soil moisture and higher VPD may induce stomatal closure in trees as a response to water stress. This, in turn, affects carbon assimilation, leading to higher $\delta^{13}$C values in tree rings and a stronger correlation between $\delta^{13}$C
and VPD (Saurer et al., 2004).

In addition to the strong, stationary hydroclimate-isotope relationships (Fig. 7 and 8), we observed significant correlations between key hydroclimate parameters and both isotopic parameters over time (Fig. 9). These findings of temporally stable and significant responses align with previous studies conducted in Central and Northern Europe (Reynolds-Henne et al., 2007; Treydte et al., 2024). Our results indicate that despite variations in climate and site conditions, three key
hydroclimate parameters - likely including water table level, precipitation, and vapor pressure deficit - remain consistently important for our $\delta^{13}$C and $\delta^{18}$O data throughout the analysed period. Contrasting findings were reported for sites in the Central Scandinavian mountains (Seftigen et al., 2011), where temporally unstable precipitation and/or temperature–$\delta^{18}$O and $\delta^{13}$C relationships were observed. These unstable relationships were associated with large-scale shifts in climate and changes in the regional precipitation pattern. The temporal stability observed in our study is particularly noteworthy because the consistency
of the hydroclimate signal registered in tree rings is crucial for potential hydroclimate reconstructions (Wilmking et al., 2020; Treydte et al., 2024). Any deviation from the assumption of temporal stability could lead to erroneous estimations of past temperature, precipitation, or water table trends, extremes, amplitudes, or drought severities in tree-ring-based reconstructions (Wilmking et al., 2020).

## 6 Summary and Outlook

Our study is the first to examine Scots pine TRW, $\delta^{13}$C and $\delta^{18}$O patterns and their responses to hydroclimate conditions in a peatland environment in Southwestern Sweden. Our findings reveal substantial differences in TRWs mean values across sites with weak to moderate relationships between site-specific chronologies. However, strong relationships in longer-term trends are observed. Notable discrepancies also exist in both mean values and long-term trends of $\delta^{13}$C. In contrast, mean $\delta^{18}$O values, site-specific chronologies, and long-term trends exhibit strong and significant similarities across all sites.

Both $\delta^{13}$C and $\delta^{18}$O records from peatland trees show robust responses to summer hydroclimate conditions, mirroring those observed at the reference site. Temperature and VPD positively affect both isotopes, while water table and precipitation show inverse relationships. Among the hydroclimate variables, water table, precipitation, and vapor pressure deficit emerge as the primary drivers across all sites. Although $\delta^{13}$C show slightly stronger response to hydroclimate conditions, $\delta^{18}$O displays a more uniform pattern, with stronger responses at the peatland sites compared to the reference site. Moreover, the temporal
stability of the responses, although significant for both isotope ratios throughout the entire analysed period, is more pronounced for $\delta^{18}$O. The weak and inconsistent responses in TRW hinder the identification of limiting growth factors. Thus, our findings underscore the value of peatland tree rings and highlight the superiority of $\delta^{13}$C and $\delta^{18}$O over TRW in dendroclimatological research conducted within such environments.

The strong responses observed in living peatland trees suggest the potential for using stable isotopes from subfossil
peatland trees to reconstruct past hydroclimatic changes, a task not achievable so far using TRW. However, merging $\delta^{13}$C or



$\delta^{18}O$ data from different sites, or combining $\delta^{13}C$ with $\delta^{18}O$ data from the same site to enhance the climate signal, requires careful consideration of long-term trends to avoid blending different low-frequency trends which could introduce long-term artifacts in hydroclimate reconstruction.

**Data availability**

The raw data supporting the conclusions of this article will be made available by the authors, without undue reservation.

**Supplement**

Supplementary data associated with this article can be found in the online version at …

**Author contributions**

KJ, SP, LF and MS conceived the study. MS, CC and JE received the funding for the study. KJ, SP, LF, JE, and MAN collected samples. KJ and MAN performed labwork. KJ analysed the data with input from KT. KJ wrote the manuscript with input from KT. VL provided water table data. SP, LF, MAN, JE, CC, VL, MS reviewed and edited the manuscript. All authors contributed to the discussion of the manuscript and approved the submitted version.

**Funding**

This study was an integrated part of the TURBERAS and MOSS projects. TURBERAS: Reconstruction of Holocene hydro-climatic fluctuations based on multi-proxy peatland records (Swiss National Science Foundation, grant no: 200021_182032). MOSS – Management strategies for tree colonized peatland ecosystems (FORMAS grant no: 2020-00936). KT acknowledges funding from the Swiss National Science Foundation (grant no: 175888).

**Acknowledgements**

The SITES and ICOS-Sweden research infrastructures are thankful for providing site-specific data from the Mycklemossen and for the accommodation during fieldwork campaigns within the Skogaryd Research Catchment.

**Conflict of interest**

The authors declare that there is no conflict of interest.

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
