# Peer review of "Peatland trees record strong and temporally stable hydroclimate information in tree-ring $\delta^{13}C$ and $\delta^{18}O$"

_EGUsphere, 2025_

## Author Response (AR1)

Answers in green

**Reviewer 1**

A very interesting and well-thought-out paper. An exemplary structure and promising conclusions. I would especially like to note line 357-358: "A key finding of our study is that the same hydroclimate variables influence both 13C/12C and 18O/16O fixation processes in tree rings." The subsequent discussion of the results (with references to other works) shows that the authors are well prepared to substantiate and defend their main conclusions.

Thank you very much for your positive and encouraging feedback. We truly appreciate your recognition of our key findings and the structure of our discussion. Please find below our responses to your two comments.

As a reviewer who liked the paper, I can make two small comments:

1. If the basic processes that are recorded by isotope ratios are identical not only in individual trees, but also in different local conditions (center of peatland, its edge, bedrock), then why is the variability of radial growth (tree-ring width) not so synchronous? I would like this issue to be given special attention in the discussion.

Thank you for this important comment. The text has been adapted as follows:

L. 355-395: In peatlands, tree growth is particularly sensitive to fluctuations in the water table, which influences root access to both water and nutrients and reduces soil aeration. Elevated water tables limit oxygen and nutrient availability, leading to environmental stress that typically manifests as narrow tree rings, while lower water tables tend to promote radial growth. (Boggie, 1972; Penttilä, 1991; Freléchoux, et al., 2000; Vitas and Erlickytë, 2007). Additionally, shallow rooting and unstable ground conditions - common in peatland ecosystems - can cause trees to tilt, triggering the formation of compression wood and eccentric growth as mechanical response to ground instability (Malinen et al., 2005; Timell, 1986). Wind and repeated ground movement may additionally alter the direction of compression wood within a tree, further distorting growth patterns (Linderholm et al., 2002; Zoltai and Pettapiece, 1974). As a result, tree rings may alternate between extremely narrow and unusually wide - varying individually among trees - reflecting the combined effects of environmental and mechanical stress. Such non-climatic influences can mask or distort the regional climate signals recorded in tree growth patterns. In contrast, mineral soil developed over bedrock provides markedly different growing conditions: it is well-drained, offers moderate to high nutrient availability, and ensures stable rooting conditions (He et al., 2025). While these characteristics may generally support more uniform growth, water availability can still be a limiting factor. The underlying granitic bedrock, although capable of storing substantial water volumes for trees (Nardini et al., 2024) due to its high porosity from weathering processes (Migoń and Lidmar-Bergström, 2001), does not necessarily provide readily accessible moisture for tree roots. This restricted access to water, despite favourable soil properties, may contribute to lower overall growth. However, due to the more homogeneous environmental conditions compared to the peatland, tree growth on mineral soil tends to be more synchronized (Linderholm et al., 2002).

Trees on the Mycklemossen peatland, similar to those in other European peatlands (Becker et al., 2008; Edvardsson et al., 2016; Holmgren et al., 2015; Smiljanić and Wilmking, 2018), grow on slightly elevated, drier hummocks (Fig. 1), which offer some protection from excessive flooding, while maintaining water access, typically fluctuating between 0 to -30 cm

in June – August (based on modelled data from 1960 to 2021). Therefore, as mentioned before, the presence of these hummocks and/or a lower water table likely contributes to wider TRW compared to the reference site, which offers comparatively drier growing conditions. However, the relatively non-synchronous growth observed between peatland sites is likely driven by at least four interacting factors: i) individualistic responses of trees to drier conditions on hummocks (Carrer, 2011), ii) variation in hummock height, leading to micro-environmental differences in growth conditions (Pouliot et al., 2011), iii) the shallower peat at the edge (~1.5 m), potentially forming a transitional zone between the central peatland (with up to 5 m of peat) and the adjacent mineral soil, and iv) localized shifts in hydrology, unstable ground conditions, and wind exposure, leading to episodes of compression wood formation, eccentric growth, and growth suppression followed by release (Linderholm et al., 2002; Malinen et al., 2005; Timell, 1986; Zoltai and Pettapiece, 1974). In addition, differences in growth patterns between the peatland and the reference site likely reflect the contrasting soil conditions: the thin mineral soil layer at the reference site (10-30 cm) offers markedly different growing conditions than the deep organic soils of the peatland (He et al., 2025). The weak to moderate coherency in growth patterns observed in our study (Fig. 2) is consistent with previous research comparing Scots pine TRWs on peatlands and mineral soils (Cedro and Lamentowicz, 2011; Hökkä et al., 2012; Linderholm et al., 2002). In comparison, a notable example comes from a study in Lithuania, where TRW chronologies from pine populations growing just a few hundred meters apart - on peatland and mineral soils - showed no significant correlation (Edvardsson et al., 2015b). In this context, the weak to moderate relationships observed in our study suggest that, while TRWs do share a common signal, it is still limited. This reinforces our assumption that tree growth across our sites is influenced by multiple co-existing environmental and site-specific factors (Edvardsson et al., 2015a, 2016; Linderholm et al., 2002). As a result of this complexity, developing chronologies that are coherent across ecologically distinct sites - or even among different locations within the same peatland - can be challenging (Smiljanić et al., 2014).

In light of the arguments and examples discussed above, we confirm that the impact of local environmental variability on stable isotope ratios is significantly reduced compared to TRW. We expanded the discussion in the manuscript to emphasize that TRW is more susceptible to micro-site hydrological variability, whereas stable isotopes more consistently reflect larger-scale climatic patterns such as temperature and precipitation.

2. Worship or fear of statistics leads to the fact that in some places the text (primarily the results) resembles cuneiform due to the abundance of numerical data and references to certain statistical estimates. This, of course, distracts from reading the text itself. I understand that these are modern requirements, but many of the presented numbers would be better either summarized in tables or even moved to the supplementary. And in general, excessive enthusiasm for statistics depresses theoretical constructions. The above does not change the good opinion of the work, which deserves to be published.

We appreciate your comment and agree that the Results section could benefit from improved clarity. We revised the section to better highlight the key findings and enhance readability, with the aim of making the main messages more accessible to the reader. As recommended by the reviewers and the editor, the values are now presented in tables (Table 1, and the tables in Fig. 2 and Fig. 3) and have been significantly removed from the main text body.

**Reviewer 2**

I found the paper interesting, very well-written and findings well-presented. The authors describe the methods and results clearly, as well as thoughtfully discuss their findings and engage with a broad literature. I fully support publishing this paper after minor revisions.

Thank you very much for your positive and encouraging feedback. We appreciate your support and will carefully address the minor revisions.

**Specific comments and recommendations:**

1. Please consider showing locations of the meteorological stations and the CRU grid on the map (Fig. 1).

We understand the suggestion to include the locations of the meteorological stations and the CRU grid in Figure 1. However, we would prefer to retain the current version of the map, as it provides a level of detail - particularly in the relief shading - that is important for illustrating the local topography of the three study sites, which are located within just 150-300 meters of each other. Adding the locations of the meteorological stations which are in 30 km distance from the study site, would require significantly expanding the map. Also, presenting the CRU grid (which has a resolution of 0.5° latitude/longitude) would require a broader spatial scale, further compromising the figure's clarity and focus.

As a compromise, we added a statement in the text indicating the relative positions of the meteorological stations (L 158-160: Kroppefjäll to the north and Såtenäs to the northeast of the study site, both ~30 km away, and the closest CRU grid point ~18 km northwest of the study site), to provide geographic context without affecting the visual resolution of the figure.

2. Could the low EPS of the TRW chronology from the EDG site be commented on, please? It is important to ensure the dating accuracy before conducting any further analysis.

Thank you for raising this important point. The dating accuracy of the TRW chronology from the EDG site has been carefully verified through standard statistical and visual cross-dating procedures and we are confident that the dating is robust.

The relatively low EPS likely results from heterogeneous growth patterns among individual trees at this site. During the design of our sampling strategy, we intentionally divided the peatland into EDG and CEN sites to investigate whether the shallower peat depth at the edge (~1.5 m) might form a transitional zone between the central peatland (with peat depths of up to 5 m) and the surrounding mineral soil - potentially affecting tree-growth dynamics. This transitional setting may explain the increased variability in growth responses, leading to lower EPS compared to the other two sites. Additionally, the observed heterogeneity in growth patterns may be linked to periods of compression wood formation and/or growth suppression followed by release in some trees, likely in response to localized shifts in hydrological conditions or competition. Although no visible signs of disturbance, such as top-kill or scars, were detected, the observed growth patterns suggest that dynamic microsite conditions may have influenced individual tree growth, not consistent across the entire tree population.

In summary, while environmental complexity at the EDG site may contribute to reduced EPS, the chronology remains reliably dated through robust visual and statistical cross-dating thanks to a number of distinct pointer years that still all trees have in common.

3. Please consider showing temperature and precipitation data over the studied period (maybe as a supplementary figure). It is important since rising temperatures are discussed as a potential reason for decreased d13C (e.g., lines 420-425).

Thank you for this suggestion. We included plots presenting the temperature and precipitation data that support the interpretation of the $\delta^{13}$C trends as a supplementary figure.

4. I find long-term trends in TRW, d13C and d18O slightly overinterpreted. The chronologies cover 60 years, which may be too short to exclude an edge-effect. Moreover, I do not find the TRW chronologies to share a strong similarity (line 385), neither do I agree that long-term trends in d13C "differ remarkably" (line 414). I would recommend moving figures with long-term trends into the supplementary.

Thank you for your comment.

Considering:

"I find long-term trends in TRW, d13C and d18O slightly overinterpreted […]. I would recommend moving figures with long-term trends into the supplementary."

We agree that referring to "long-term trends" in the context of tree-ring chronologies spanning approximately 60 years may be somewhat exaggerated. We believe, however, that presenting "longer-term trends" -or more accurately, "multi-decadal trends" - remains valuable, as such patterns are often overlooked in tree-ring studies focused on climate reconstructions. Although our study does not aim to perform a climate reconstruction per se, showing low-pass filtered data in the manuscript allows highlighting these broader trends, which are particularly relevant when integrating TRW and/or isotope chronologies from ecologically distinct sites. Therefore, while we propose to retain the concept of longer-term trends in the manuscript, we suggest to simply call it: "low-pass filtered data" or "multi-decadal trends", to avoid overinterpretation. We remain convinced that including these trends is important for providing a robust foundation for climate reconstruction and merits discussion in the manuscript.

Considering:

"The chronologies cover 60 years, which may be too short to exclude an edge-effect."

We are not entirely certain what is meant by "edge-effect" in this context, but we assume the comment rather may refer to the "age effect" (i.e., juvenile growth-related trends). If that is the case, we would like to kindly clarify a few points.

We acknowledge that a negative trend is visible in TRW series at the CEN site. While this may reflect a juvenile effect in some cases, it is also likely that the presence of compression wood and resulting eccentric growth - common under the unstable and waterlogged conditions typical of peatlands - has contributed to this pattern.

This possibility - that the observed negative trends may be influenced more by compression wood and eccentric growth than by juvenile effects - is further supported by the fact that many of the trees included in the chronologies are older, typically around 80-100 years. Due to sampling limitations, we often did not reach the pith, meaning that the actual age of the trees exceeds the length of the measured series, although we did not apply pith-offset

estimations. In some cases, we also excluded innermost rings during measurement or removed them during cross-dating, as they did not cross-date reliably.

Finally, all tree-ring series were detrended under the assumption that non-climatic influences - including potential age-related effects - were effectively removed. We therefore consider the multi-decadal trends presented in our study to be robust and meaningful.

Regarding stable isotope trends, we consider no effect of age-related trends because for carbon such juvenile trends would be restricted to the first few decades of the tree life which we excluded from the analysis. The same holds for oxygen which seem less affected by age-related trends anyway (Klesse et al., 2018).

Considering:

"Moreover, I do not find the TRW chronologies to share a strong similarity (line 385), […] neither do I agree that long-term trends in $\delta^{13}$C 'differ remarkably' (line 414)."

We agree that the TRW chronologies do not exhibit a strong similarity, but rather a moderate one. Likewise, while the long-term trends in $\delta^{13}$C do not differ remarkably, there are some visible discrepancies.

Therefore, we propose replacing the word "strong" with "moderate" in L 385 (now L 396) and adjusting "differ remarkably" to "differ slightly" in L 414 (now L 426).

5. I agree with the Referee#1 that Results are over saturated with numbers and statistics. No doubts they are important, but please consider summarising them to facilitate a smooth understanding of their main message.

We appreciate your and reviewer #1's comment and agree that the Results section could benefit from improved clarity. We revised the section to better highlight the key findings and enhance readability, with the aim of making the main messages more accessible to the reader. As recommended by the reviewers and the editor, the values are now presented in tables (Table 1, and the tables in Fig. 2 and Fig. 3) and have been significantly removed from the main text body.

6. The d13C values suggest that pines at the reference site are drought stressed (lines 400-405). This finding is independently supported by a positive correlation between TRWs and precipitation (see Fig. 6). The authors may consider mentioning this in the text.

Thank you for this insightful observation, we agree. We revised the text accordingly to explicitly mention and integrate this supporting evidence. The text has been adapted as follows:

L. 411-417: Trees in the Mycklemossen peatland, however, grow on hummocks, with convenient yet not excessive access to water pools (Fig. 1) and shallow, horizontally extended root systems (>- 20 cm, Heikurainen, 1955; He et al., 2023) (Edvardsson et al., 2012b). Therefore, stomatal conductance of these trees may remain high, leading to lower $\delta^{13}$C values. In contrast, higher $\delta^{13}$C values (Fig. 3) observed alongside narrower tree rings (Fig. 2) at the reference site underscore its dependence on soil moisture and indicate drought stress (Leavitt, 1993; Saurer et al., 1995; Treydte et al., 2014). The highest $\delta^{13}$C values recorded at this driest site suggest reduced stomatal conductance and increased water-use efficiency (Gessler et al., 2014; Saurer et al., 1995, 2014; Treydte et al., 2001).

7. Please consider discussing similarities and differences between the sites: in particular, is the middle site (EDG) closer to the central peatland or to the mineral soil site? Does this relationship change across the tree-ring parameters?

The text has been adapted as follows:

L 114-123: Although trees at the CEN and EDG sites grow on slightly elevated, drier hummocks, they remain rooted in deep organic soils – up to 5 m in the center and approximately 1.5 me deep at the edge. Despite these micro-elevational differences, both sites are still relatively moist due to the proximity of trees to water pools (Fig. 1). In contrast, the reference site is located on bedrock and features a well-drained mineral soil layer of 10-30 cm thickness, leading to distinctly different hydrological conditions, thus, classified as a dry site. The reference site is assumed to be comparable to other, ecologically non-extreme, temperate sites documented in the literature (Treydte et al., 2007, 2024), where mixed climate signals are recorded in TRW. This comparison allows us to test whether site conditions modulate TRW, $\delta^{13}$C, and $\delta^{18}$O variability and their responses to various hydroclimate variables. In addition to hydrology, the sites also differ in pH and nutrient availability. Trees at the CEN and EDG sites experience lower pH levels and rely solely on nutrients from precipitation. Meanwhile, trees at the REF site benefit from access to both mineral soil nutrients and atmospheric deposition.

8. The interpretation of positive relationship between TRW and d18O at the peatland site (lines 440-450) appears unclear to me and would benefit from re-formulating. In particular, the paragraph starts with stating that higher d18O can be interpreted as drier conditions that in turn, promote tree growth. However, this interpretation is later discarded, and I struggle to understand the reasons.

Thank you for this comment. We reformulated the paragraph as follows:

L 456-468: The observed positive relationship between detrended TRW and raw $\delta^{18}$O chronologies at the central peatland site initially suggests that slightly lowered water tables and relatively drier surface conditions on hummocks may promote tree growth (Edvardsson et al., 2016; Smiljanić and Wilmking, 2018). Under such conditions, trees may benefit from improved soil aeration while still accessing sufficient moisture, enabling continued stomatal conductance and carbon uptake even under atmospherically dry conditions (i.e., high VPD). Importantly, higher $\delta^{18}$O values do not necessarily indicate soil water limitation or drought stress, but rather enhanced leaf-level evaporative enrichment driven by high atmospheric water demand. When soil moisture is adequate, stomatal conductance will remain high also under high VPD, enabling both continued transpiration and xylem cell production. Therefore, the observed positive relationship between $\delta^{18}$O and TRW likely reflects conditions where trees experience high evaporative demand but no major constraints on water uptake, supporting both isotopic enrichment and radial growth. The strength of the relationships between TRW, $\delta^{13}$C, and $\delta^{18}$O, however, varies across sites (Fig. 4 and Fig. 5), ranging from strong to weak or non-significant. This underscores the complex interplay between local hydrology and atmospheric conditions and highlights that wood production in peatland pines cannot be inferred from stomatal behaviour alone, but must be understood within site-specific environmental contexts.

**Technical corrections:**

1. line 40 - recorder --> recorded

Corrected

2. lines 67-69 need a citation

Citations were added:

Barbour, M.M., Roden, J.S., Farquhar, G.D. et al. Expressing leaf water and cellulose oxygen isotope ratios as enrichment above source water reveals evidence of a Péclet effect. Oecologia 138, 426–435 (2004).

Roden, J.S., Lin, G. and Ehleringer, J.R., 2000. A mechanistic model for interpretation of hydrogen and oxygen isotope ratios in tree-ring cellulose. Geochimica et Cosmochimica Acta, 64(1), pp.21-35.

3. line 100 lacks a full stop

Corrected

4. line 110 with water pools --> to water pools

Corrected

5. please consider moving the lines 115-120 to the introduction

We agree with the suggestion; thus, we moved the content from L 115-120 to the introduction (L 82-86) to improve the contextual framing of the study.

6. line 158 - "similar relationships were observed between isotope chronologies" is unclear in the context of comparing climate parameters

We agree the original sentence was unclear. It has been revised to:

L 162-163: Additionally, stable isotope chronologies were correlated separately with both gridded and instrumental hydroclimate data, revealing similar relationships in each case (analyses conducted but not shown).

7. line 176 - consider explaining what is delta t

The explanation was added to the text:

L 179-183: To address this, a delta T correction - an adjustment for systematic temperature differences between stations during the overlapping period was applied to harmonize the data from both stations, resulting in a homogeneous weather dataset (r-square > 0.90 for all parameters during the overlapping period).

8. line 198 - visual comparison cannot be conducted by calculating correlation coefficients

Corrected

9. The Results section is written in the past tense, whereas usually the past tense is used for Methods only - please consider changing.

Thank you for this observation. While we understand that some disciplines prefer the present tense for the Results section, the use of past tense is also common in environmental and

paleoclimatic research when referring to specific findings from the dataset analyzed. For consistency and clarity, we retained the past tense in the Results section, after consultation with the editor.

10. lines 217-218 are hard to grasp - please consider reformulating

Following Your and Reviewer #1's comments, we have revised the Results section - particularly the statistical descriptions - to improve clarity and better highlight the key findings.

11. Table 1 - please explain Rbar, EPS, and AR1 in the caption

Corrected

12. Figure 2 - please consider using the same scale for the plots with raw TRW measurements

We appreciate the reviewer's suggestion. We intentionally used different y-axis scales in the three panels showing raw TRW measurements to accurately represent the growth levels at each site. If we were to apply a uniform scale - based on the widest TRWs from the CEN or EDG sites - the narrower values from the REF site would appear nearly flattened, making it difficult to assess and compare growth rates meaningfully. For this reason, we prefer to retain the site-specific scales. To avoid confusion, we added a short note to the figure caption indicating that the y-axis scales differ between panels.

13. line 316 "temperature whereas" --> "temperature, whereas"

Corrected

14. The section 4.3 refers to the figures 5a and 5b, but it should be 4a and 4b.

Corrected

15. The section 4.2 refers to the figures 6a and 6b, but it should be 5a and 5b.

Corrected

16. line 479 should refer to the Fig. 3 instead of Fig. 2

Corrected

17. lines 482 and 528 Central and Northern Europe --> central and northern Europe

Corrected

18. line 493 - please remove "Obviously"

Corrected